# The Role and Diagnostic Potential of Insulin-like Growth Factor 1 in Diabetic Retinopathy and Diabetic Macular Edema

**DOI:** 10.3390/ijms26093961

**Published:** 2025-04-22

**Authors:** Akanksha Malepati, Maria B. Grant

**Affiliations:** 1UAB Heersink School of Medicine, University of Alabama at Birmingham, Birmingham, AL 35233, USA; amalepat@uab.edu; 2Department of Ophthalmology and Visual Sciences, University of Alabama at Birmingham, Birmingham, AL 35233, USA

**Keywords:** diabetes, diabetic macular edema, diabetic retinopathy, insulin-like growth factor 1, growth hormone, inflammation

## Abstract

Diabetes mellitus (DM) is a chronic metabolic disorder that results in hyperglycemia, leading to multiple microvascular and macrovascular complications, including significant ocular damage resulting in the development of diabetic retinopathy (DR) and diabetic macular edema (DME). Many factors contribute to the pathogenesis of DR and DME, including hyperglycemia-mediated vascular and neuronal abnormalities and local and systemic inflammation. Growth hormone (GH) and insulin-like growth factor-1 (IGF-1) have been implicated in the initiation and progression of DR and DME through a variety of mechanistic processes. In this review, we provide a comprehensive synopsis of the diverse roles and molecular pathways supporting IGF-1 in the pathogenesis of DR and DME, elucidating its range of effects from detrimental to protective, depending on the context and stage of disease. We further investigate the underlying inflammatory processes regulated by IGF-1 and examine how the interaction of IGF-1 with key signaling molecules influences these inflammatory mechanisms. Additionally, the potential of serum IGF-1 as a biomarker for the progression of DR and DME in clinical practice is discussed. Finally, we consider current therapeutic approaches for DR and DME in relation to IGF-1 and explore novel therapeutic targets and innovative delivery methods. By providing an in-depth understanding of IGF-1’s role in the pathogenesis and progression of DR and DME, this review underscores the diagnostic utility of serum IGF-1 and puts forth new treatment strategies to improve the management of DR and DME.

## 1. The Diabetes Mellitus Epidemic

Diabetes mellitus (DM) is a debilitating disorder that currently affects 11.6% of the United States (US) population [1]. Notably, 38.4 million individuals are afflicted with DM, with 29.7 million being diagnosed and 8.7 million being undiagnosed. The prevalence of total DM has significantly increased in adults within the past twenty years, thus leading to a global health crisis [1,2]. It is estimated that the total cost of diagnosed diabetes in the US is USD 412.9 billion, consequently establishing both its economic and personal burden [3].

In general, DM is a metabolic disease characterized by persistent hyperglycemia. DM can be systematically classified into several distinct types, including Type 1 diabetes mellitus (T1D), Type 2 diabetes mellitus (T2D), maturity-onset diabetes of the young (MODY), gestational diabetes, neonatal diabetes, diseases of the exocrine pancreas (Type 3c diabetes), endocrinopathies, and drug-induced diabetes [4]. T1D and T2D are the most prevalent forms of DM. T1D is characterized by a lack of insulin secretion due to the autoimmune destruction of pancreatic β cells, most commonly presenting in children and adolescents. T2D is characterized by insufficient insulin action due to pancreatic β cell dysfunction or insulin resistance, most commonly presenting in adults [5,6]. A multitude of risk factors are linked to the pathogenesis of T2D, including increased body weight, poor diet, sedentary lifestyle, family history, and other metabolic syndrome components [7]. Ultimately, DM leads to several complications, such as cardiovascular disease, stroke, peripheral vascular disease, neuropathy, end-stage renal disease, and retinopathy [8,9]. Therefore, continued research in this area is essential to improving health outcomes and population health.

## 2. Ocular Complications of Diabetes

DM can lead to several ocular complications, such as diabetic retinopathy (DR), diabetic macular edema (DME), diabetic papillopathy, glaucoma, and cataract [10]. DR is a leading cause of blindness among working-age adults in the US. An estimated 9.6 million individuals, or 26.4% of those with DM, have DR [11]. Remarkably, over 75% of patients diagnosed with T1D and over 25% of patients diagnosed with T2D for more than twenty years have some degree of DR [12].

DR is the most common microvascular manifestation of DM that is caused by prolonged hyperglycemia, vascular endothelial dysfunction, and retinal neurodegeneration resulting from neurovascular disruptions. DR can be classified into two types: non-proliferative diabetic retinopathy (NPDR) and proliferative diabetic retinopathy (PDR). Because of hyperglycemia-induced damage to the retinal vasculature, outpouching of the vessel walls, known as microaneurysms, occurs. These microaneurysms can rupture and cause intraretinal hemorrhages. Furthermore, weakened vessel walls and breakdown of the blood–retinal barrier (BRB) can lead to leakage of fluid and lipids and their accumulation in the retina and, more specifically, the macula. This results in a condition known as DME, which is a cause of vision loss. As these fluid deposits resolve, they leave behind hard lipid exudates. Eventually, NPDR causes the obstruction of vessels and infarction of the retinal nerve fiber layer, thus leading to cotton-wool spots. NPDR can be subclassified into four stages: mild, moderate, severe, and very severe. These stages are distinguished by specific clinical features, such as the extent of microaneurysms, intraretinal hemorrhages, lipid exudates, intraretinal microvascular abnormalities (IRMAs), and cotton-wool spots, which are focal infarcts of the retinal nerve fiber layer. If ischemia is sustained, aberrant retinal vasculature or neovascularization develops; this stage marks the transition point to PDR. These new vessels are often leaky due to a lack of tight junctions, are structurally weak, and are prone to hemorrhage, thus potentially leading to vision loss. The neovascularization process combined with the fibrosis of new vessels can ultimately result in tractional retinal tears and detachment [10,12,13,14,15,16]. Consequently, early detection and comprehensive management of DM are essential to prevent the progression of DR and DME and to preserve visual function in patients.

## 3. Pathogenesis and the Role of Insulin-like Growth Factor 1 (IGF-1) in DR and DME

The pathogenesis of DR and DME is thought to be multifactorial, involving metabolic pathways, vascular abnormalities that lead to angiogenesis, inflammation, and retinal neurodegeneration [17]. The role of IGF-1 in these processes is incompletely understood. IGF-1 and IGF-1 receptors are expressed throughout the retina on endothelial and neuronal cells. Additionally, the expression of IGF-1 and IGF-1 receptors is modulated by hypoxic and hyperglycemic conditions [18].

IGF-1 has been implicated in the induction of neovascularization via the elevation of vascular endothelial growth factor (VEGF)-driven endothelial cell proliferation [19,20,21], as IGF-1 is needed for VEGF expression. Vascular endothelial cell IGF-1 receptor knockout mice have reduced retinal neovascularization [22]. Specifically, IGF-1 increases VEGF expression via a PI-3K/Akt-dependent mechanism involving HIF-1α and NF-κB/AP-1. PI-3K/Akt activation stabilizes HIF-1α and activates NF-κB/AP-1, which then binds to hypoxia-response elements in the VEGF promoter region, ultimately leading to the upregulation of VEGF and angiogenesis [23]. IGF-1 signaling further enhances retinal angiogenesis by activating JNK/AP-1, partially via the PI-3K/Akt pathway, thereby amplifying retinal VEGF expression [23]. Additionally, previous studies have shown that the destabilization of retinal neovasculature is driven by lysophosphatidic acid (LPA) [24]. However, IGF-1 stabilizes VEGF-derived neovasculature by prolonging the activation of extracellular signal-regulated kinase (ERK), which antagonizes LPA-dependent regression of retinal neovessels via inhibition of the Rho/ROCK pathway [24]. Under physiological conditions, these studies display the proangiogenic effect of IGF-1 and its contribution to the development of aberrant vessels within the retina and DR.

The blood–retinal barrier (BRB) is a crucial structure that regulates the microenvironment of the retina and protects the retinal vasculature [25]. The inner blood–retinal barrier (iBRB) consists of retinal capillary endothelial cells with tight junctions in between. These endothelial cells are enveloped by pericytes and the foot processes of Müller cells [26]. The function of the iBRB is to maintain retinal homeostasis by regulating the transport of molecules across the retinal vasculature and protecting against toxins [27]. The outer blood–retinal barrier (oBRB) is composed of retinal pigment epithelium. The oBRB is responsible for regulating transport between the retina and choriocapillaris [26]. The breakdown of the iBRB, resulting in increased vascular permeability, is a hallmark of the progression of DR and DME [28]. Claudin isoforms 1 and 5 and occludin are present in tight junctions of retinal vasculature [29]. High intraocular IGF-1 triggers the breakdown of the BRB and increase paracellular transport. This effect is mediated through decreased expression of claudin-1 and maintained expression of claudin-5 and occludin [30]. In addition to these molecular changes, the overexpression of IGF-1 in normoglycemic transgenic murine eyes induces several pathophysiological changes that resemble DR. These murine retinas exhibit progression from NPDR to PDR to retinal detachments. Additional changes include vascular basement membrane thickening, pericyte dropout, microvascular abnormalities, venule dilation, and neovascularization in the retina and vitreous cavity [31]. Furthermore, in hyperglycemic conditions, IGF-1 upregulation leads to a signaling switch, where Src homology 2 domain-containing protein tyrosine phosphatase substrate 1 (SHPS-1) is phosphorylated and associates with integrin-associated protein (IAP), resulting in aberrant activation of the Akt and MAPK pathway [32,33]. This leads to endothelial cell dedifferentiation and aberrant growth, contributing to increased vascular permeability [34,35]. Indeed, the inhibition of SHPS phosphorylation and the SHPS-1/IAP association led to suppression of the Akt pathway, decreased VEGF synthesis, and modulation of IGF-1 signaling, ultimately reducing vascular permeability [36]. Overall, these changes in the BRB lead to increased vascular permeability and progression of DR and DME, thus demonstrating the pivotal role of IGF-1. Figure 1 presents a summary of the pathological role of IGF-1 in the progression of DR and DME.

In spite of IGF-1’s role in inducing the VEGF pathway, a recent study has shown that IGF-1 and dopamine work synergistically to lower VEGF 1 receptor expression. Therefore, adequate availability of dopamine within the retinal microenvironment can prevent the pathologic effects of IGF-1, thus leading to reduced angiogenesis in PDR [37]. Moreover, in PDR, endothelial cell dysfunction is a critical pathological feature that disrupts normal retinal hemodynamics. This dysfunction arises from decreased bioavailability of nitric oxide, which normally regulates blood flow, alleviates retinal ischemia, and promotes the endothelial cell proliferation and migration necessary for endothelial repair [38]. Despite reduced nitric oxide availability, the IGF-1 receptor has been shown to be associated with increased in situ endothelium regeneration [39]. These findings demonstrate the broad role that IGF-1 plays in neovascularization and endothelium repair.

DR is associated with the cell death of several retinal cells, including endothelial cells, pericytes, glial cells, and ganglion cells [40]. In diabetic rat retinas, markers of apoptosis, including the quantity of TUNEL-immunoreactive cells and cells positive for phospho-Akt, caspase-3, and Bad, were significantly increased across all layers of the retina, including the photoreceptor layer, inner nuclear layer (INL), and ganglion cell layer, compared to IGF-1-treated diabetic rat retinas. However, systemic treatment with IGF-1 did not prevent hyperglycemia in diabetic rats. Therefore, these results demonstrate that early treatment of DR with systemic IGF-1 has been shown to reduce apoptotic markers and, subsequently, retinal cell death, even in the setting of persistent hyperglycemia [41]. This effect occurs because IGF-1 signaling provides neuroprotection against apoptosis in retinal neurons via modulation of the PI-3K/Akt pathway and the inhibition of caspase-3 activity [41,42]. IGF-1 is also essential for photoreceptor survival, as it conserves photoreceptor structure and function and regulates retinal metabolism to sustain the high metabolic and glycolytic energy requirements of photoreceptor cells [43]. Additionally, somatostatin (SST) is a neuroprotective factor synthesized by the retina, and its levels decrease concurrently with the progression of neurodegeneration in DR. It has been shown that SST protects against apoptosis by augmenting IGF-1-mediated Akt phosphorylation [44]. These findings demonstrate the necessity of IGF-1 in the context of neuroprotection and cell survival.

In the setting of retinal injury and regeneration in zebrafish, IGF-1, alongside insulin and fibroblast growth factor, is a necessary signaling component to reprogram and activate Müller glia. Müller glia can transition to multipotent progenitors that respond to retinal injury and initiate retinal repair supporting the potential role of IGF-1 in retinal regeneration in humans [45,46]. Moreover, IGF-1 is responsible for regulating the degree of neurotrophic molecules such as IL-4 and brain-derived neurotrophic factor (BDNF) [47]. IL-4 increases retinal cell survival and proliferation, while BDNF inhibits retinal cell proliferation [48]. However, equally important is understanding the signaling pathways involved in IGF-1-mediated retinal cell proliferation include IGF-1 receptor endocytosis, which stimulates the MAPK/ERK and JNK signaling pathways. Furthermore, this process activates additional signaling pathways, including Src, PI-3K, protein kinase 3 C delta, and phospholipase C, alongside matrix metalloproteinases and EGFR, which collectively lead to retinal cell proliferation [47]. Ultimately, understanding the homeostatic role of IGF-1 in retinal cell survival and repair highlights its potential benefits in managing diabetic retinopathy. Figure 2 provides an overview of the protective role that IGF-1 may play in the progression of DR and DME.

## 4. The Effects of IGF-1 on Inflammatory Processes in DR and DME

Retinal inflammation is one of the causal factors of DR [49]. Notably, leukocyte adhesion to retinal vasculature and resulting leukostasis contributes to BRB breakdown, lack of capillary perfusion, and endothelial cell damage and death [50]. This process is partially mediated by intracellular adhesion molecule-1 (ICAM-1), which facilitates the migration of leukocytes from the circulation into the retina. ICAM-1 also binds to β2 integrin lymphocyte function-associated antigen (LFA-1), leading to the release of pro-inflammatory cytokines, including interleukin (IL)-1, IL-6, and tumor necrosis factor-α, as well as matrix metalloproteinases and chemokines [51]. ICAM-1 expression is upregulated in the presence of VEGF and local IGF-1 [30,52]; high intraocular IGF-1 levels have also been shown to increase the recruitment of bone marrow-derived inflammatory cells [53]. These factors initiate the processes that lead to a chronic inflammatory state within the retina, directly contributing to the pathogenesis of DR.

Furthermore, microglia are tissue-resident monocytes that inhabit the retina. There are two main phenotypes of microglia, classically referred to as M1 and M2. M1 microglia are responsible for inflammatory processes and neurotoxicity by releasing pro-inflammatory cytokines such as TNF-α, IL-1β, IL-6, and IL-8. Conversely, M2 microglia contribute to anti-inflammatory processes and neuroprotection by releasing anti-inflammatory cytokines such as TGF-β, IL-4, IL-10, and IL-13 [53]. In the context of a normal retinal environment, microglia play a role in establishing normal vasculature and myelination [54]. Additionally, when inflammation is required to clear immunologic threats in the retina, the M1 phenotype is activated. Once the threat has been neutralized, M1 microglia revert to a surveillance state, and the M2 phenotype is activated to initiate repair and clear debris in order to maintain homeostasis. This dichotomized classification has faced criticism in recent years, as advancements in transcriptomics and proteomics have shown the heterogeneity of microglia, thus leading to a more nuanced classification system, and now it is believed that cells transition between an M1 and an M2 state and express markers of both types at the same time [55].

During DR, these homeostatic mechanisms are dysfunctional. Initially, in the diabetic retina, the upregulation of cytokines and growth factors leads to neuroprotection and the maintenance of retinal function [56]. However, the long-term activation of cytokine-mediated inflammatory responses leads to the expansion of microglia. Microglia are significantly increased in number and hypertrophic, with clustering around retinal vasculature, microaneurysms, intraretinal hemorrhages, cotton-wool spots, optic nerve, and novel vessels in DR and DME [57]. Significantly, in the context of solely DR, microglia are found in the inner retinal layers, while in DME, microglia have been shown to infiltrate both into the subretinal space and outer retina [57]. These microglia display an M2 phenotype in the early stages of DR and then switch to an M1 phenotype in the advanced stage, which has been shown to correlate to vision loss [58]. The switch to an M1 phenotype is also associated with hypoxia-induced neovascularization, a stage of PDR [59]. Furthermore, in addition to pro-inflammatory cytokines, retinal microglia release complement proteins and regulators, such as C3d [60], C5a [61], C5b-9 [62], and the membrane attack complex (MAC) [62]. However, retinal microglia do not release C-reactive protein (CRP), mannose-binding lectin (MBL), C1q, or C4, indicating that complement activation is independent of a C4-dependent pathway. Pro-inflammatory cytokines released by M1 microglia are upregulated in diabetic patients and have been positively correlated with the severity of DR [63]. Consequently, this chronic inflammatory state leads to a futile cycle and progression of DR and DME [64].

With respect to the relationship between IGF-1 and retinal microglia, intraocular IGF-1 expression is upregulated in retinas with marked gliosis [30]. Additionally, IGF-1 has been shown to be mitogenic for microglia [65]. In mice that overexpress the IGF-1 receptor, the extensive gliosis and microgliosis leads to increased TNF-α and CCL-2, which is correlated with vision loss, characterized by decreased retinal layer thickness, the destruction of retinal neurons, specifically rod photoreceptors, and heightened oxidative stress [66]. Conversely, a complete deficiency of IGF-1 has been shown to cause absent ERG amplitudes, breakdown of synaptic terminals, and vision loss [67]. Thus, the expression of IGF-1 requires careful regulation. Overall, IGF-1 likely impacts retinal inflammation in DR and DME in a context-dependent manner, and the fine-tuning of IGF-1 expression is necessary to prevent the progression of DR and DME. Figure 3 shows the role of IGF-1 in the inflammatory processes contributing to DR and DME.

There are a multitude of factors that work in concert with IGF-1 in the context of the inflammatory processes of DR and DME. TGF-β is a pleiotropic immunoregulatory cytokine. TGF-β is implicated in the maintenance of microglial cells in a quiescent state, thereby reducing neuroinflammation [68]. Indeed, the deletion of TGF-β signaling has been shown to exacerbate neurodegeneration and contribute to neovascularization in murine models [69]. Furthermore, the loss of ocular TGF-β has been shown to induce a retinal phenotype that resembles PDR, with microaneurysms, leaky capillaries, retinal hemorrhages, and a loss of differentiated pericytes [69]. Interestingly, microglia located in hypoxic areas of the retina express lower TGF-β and elevated IGF-1, thus promoting angiogenesis. However, supplementation with TGF-β1 in hypoxic conditions has been shown to regulate IGF-1 levels in microglia, thereby reducing neovascularization. Additionally, TGF-β modulates retinal microglia to decrease the production of chemoattractant factors, thus reducing monocyte recruitment and leukostasis and consequently mitigating pathologic neovascularization. Therefore, under hypoxic conditions, TGF-β signaling functions independently of VEGF-1 regulation to maintain microglial homeostasis and suppress the expression of pro-angiogenic IGF-1 in retinal microglia, thus leading to decreased leukostasis and retinal neovascularization [70]. Additionally, estradiol and IGF-1 have been shown to act synergistically to induce neuroprotection and neurogenesis via common signaling pathways such as MAPK and PI-3K/Akt [71,72]. The inhibition of glycogen synthase kinase 3β (GSK3β), a downstream component of the PI-3K/Akt signaling pathway, is hypothesized to be a critical element in the neuroprotective mechanism [73]. Therefore, the tight regulation of estradiol in relation to IGF-1 may have the potential to protect against the progression of DR and DME. Finally, angiotensin II has been shown to induce an increase in M1-phenotype microglia, but when angiotensin II is co-administered with IGF-1, the M1 phenotype is suppressed, and the M2 phenotype is favored [74]. Within the setting of DR, further understanding of these factors’ role in modulating IGF-1 can provide significant new therapeutic insights.

## 5. Systemic Effects and Clinical Findings of IGF-1 on DR and DME

High systemic IGF-1 has been shown to contribute to the breakdown of the BRB and increase paracellular transport [30]. As such, understanding the relationship between systemic IGF-1 and the progression of DR and DME remains essential. However, this relationship between serum IGF-1 levels and DR in clinical settings remains a subject of ongoing debate. Historically, cases of pituitary infarction have demonstrated regression of PDR in some patients, thereby implicating growth hormone (GH) and IGF-1 in the progression of DR [75]. Similarly, pituitary ablation has also been associated with the regression of DR, thus providing additional evidence for the role of IGF-1 in DR progression [76]. Interestingly, in GH-deficient subjects, microvascular complications of DR were absent, thus supporting that GH and IGF-1 promote vascular abnormalities in DR [77].

Contemporary clinical studies have also been conducted to further examine the relationship between serum IGF-1 and the progression of DR and DME. One study has shown that in individuals over the age of 30, higher levels of IGF-1 were associated with an increased prevalence of PDR, regardless of the status of insulin usage. In individuals not using insulin, elevated IGF-1 levels were also linked to a higher prevalence of moderate NPDR [78]. Additionally, as individuals transition from post-puberty to elderly age, serum IGF-1 declines in a gradual non-linear manner, with no differences between sexes [79]. Consequently, younger patients with DM had a higher prevalence of PDR when compared to elderly patients with DM for the same duration, which may be attributed to the higher serum IGF-1 levels within the younger group [80]. Supporting this, one prospective study found that despite intravitreal aflibercept treatment, patients with PDR had significantly higher systemic IGF-1 levels compared to those with NPDR, reinforcing IGF-1’s role in DR progression independent of VEGF inhibition [81]. These studies display the potential detrimental effects of high serum IGF-1 on DR.

Despite these findings, several studies have also demonstrated a protective role of serum IGF-1. Pregnant patients with DR have lower levels of serum IGF-1 and IGFBP-3 compared to pregnant patients without DR, thus illustrating a protective effect for IGF-1 [82]. A decline in serum IGF-1 levels has been associated with the progression of diabetic microvascular complications, including DR [83]. The severity of DR in patients with T1D was inversely correlated to serum IGF-1 levels resulting in the notion of utilizing low IGF-1 levels as a clinical marker for closer management of DR [84]. A longitudinal study in children and adolescents with Type 1 diabetes demonstrated similar findings, showing that low mean IGF-1 levels were independently associated with the progression of DR [85]. The patients referenced in these studies did not exhibit abnormally low serum IGF-1 levels, which is indicative of hypopituitarism or growth-hormone deficiency [86]. Therefore, these observed declines in serum IGF-1 likely reflect the progression of DR rather than an underlying GH deficiency.

One large cross-sectional study has also shown no association between serum IGF-1 concentrations and progression of DR [87]. Overall, these conflicting findings highlight the complex relationship between serum IGF-1 and DR and underscore the need for further research to elucidate this connection and explore the potential of IGF-1 as a future diagnostic tool.

## 6. Therapeutic Potential of Targeting IGF-1

Intravitreal anti-VEGF treatment injections are used for the treatment of DME to antagonize VEGF in the retina, thus leading to decreased development of neovascularization and vascular leakage [88]. In recent years, anti-VEGF agents have been preferred over the use of laser photocoagulation for the management of DME, as anti-VEGF has been shown to improve visual and anatomic outcomes to a greater extent while preventing laser-therapy-induced ocular complications [89]. However, the frequency of anti-VEGF injections proves to be burdensome to patients as several are lost to follow-up [90]. Furthermore, a substantial proportion of patients are refractory to anti-VEGF therapy [91]. Indeed, the RISE/RIDE Phase III trial of monthly intravitreal ranibizumab injections showed the persistence of DME and increased central foveal thickness in approximately 23% of patients [92]. Though intravitreal corticosteroid injections have been used for refractory patients, corticosteroids have a lower safety profile and can lead to the formation of cataracts and ocular hypertension [93]. Therefore, the development of alternative therapies is necessary to address the needs of this population of patients and improve both clinical outcomes and quality of life.

The GH/IGF-1 axis is regulated by the secretion of GH from the anterior pituitary gland, which is stimulated by growth hormone-releasing hormone (GHRH) released from the hypothalamus. Subsequently, IGF-1 is produced and released, primarily from the liver in response to GH stimulation. IGF-1 then exerts its effects by binding to IGF-1 receptors that are distributed throughout various body tissues [94]. Increased levels of pituitary GH contribute to the pathogenesis of DR [95], specifically PDR [96]. This is further supported by acromegaly patients with excessive GH/IGF-1 levels showing increased retinal vessels and vascular branching points despite the absence of diabetes [97]. Somatostatin (SST) is a cyclic hormonal peptide that modulates the GH/IGF-1 axis. Specifically, SST lowers circulating GH while normalizing IGF-1 levels [98]. Consequently, the SST analog Octreotide is a drug that is capable of stabilizing the BRB and preventing angiogenesis in DR and DME [99]. SST has been shown to increase the activity of tyrosine phosphatases SHP1 and SHP2 by binding to its somatostatin receptors (Sst1-5), thus inhibiting receptor tyrosine kinase signal transduction and leading to the prevention of the angiogenic effects of growth factors such as IGF-1 and VEGF. Additionally, the anti-angiogenic effect occurs through the antagonism of the GH-IGF-1 axis [100]. Figure 4 shows a representation of the hypothalamic–pituitary–somatotropic axis.

Prior to the development of intravitreal anti-VEGF treatment, SST analogs were tested to assess their therapeutic potential for DR. SST analogs were found to stabilize severe PDR [101], delay the time to necessity of panretinal photocoagulation in advanced DR patients, reduce vitreal hemorrhages in PDR patients following full scatter laser coagulation, repress new bleeding, and halt vision loss in patients with failed photocoagulation therapy [102,103,104]. However, the peripheral delivery of SST leads to systemic side effects, such as gastritis, damage to the superficial and deep layers of gastric mucosa, focal gastric mucosal atrophy [105], and detrimental changes in glucose homeostasis [106]. Because of these systemic effects, new methods of delivery for SST analogs are currently being tested. For example, topically administered SST has been shown to decrease neurodegeneration and decrease activation of M1 microglia in DR [107]. The administration of SST with magnetic nanoparticles via intraocular injections is also currently being investigated [108]. Overall, further research into novel methods to centrally deliver this drug for the regulation of the GH/IGF-1 axis should be evaluated as a potential alternative therapy for DR and DME.

## 7. Conclusions

IGF-1 has a critical and multifaceted role in the pathogenesis of DR and DME, which is summarized in Table 1. In certain contexts, IGF-1 has been implicated in retinal neovascularization and increased vascular permeability, which are key pathological features of DR and DME. Furthermore, IGF-1 significantly contributes to retinal neuroinflammation by inducing leukostasis, activating pro-inflammatory microglia, and releasing pro-inflammatory cytokines. Paradoxically, however, IGF-1 is also crucial for endothelium regeneration, reducing expression of the VEGF receptor, enhancing cell survival, and initiating retinal repair following injury. Consequently, IGF-1’s role in the setting of DR and DME is complex, ranging from neuroprotective effects to the promotion of neuroinflammation and microvascular complications.

Clinical studies investigating the relationship between serum IGF-1 levels and the progression of DR and DME have yielded conflicting results, underscoring the need for further research to elucidate the relationship between the two. Delineating the relationship between serum IGF-1 levels and the stages of DR and DME progression could establish serum IGF-1 as a valuable biomarker for these conditions. Further research should focus on characterizing the relationship between retinal inflammation and serum IGF-1 levels, given IGF-1’s involvement in neuroinflammation. This approach could provide a new method for monitoring the progression of ocular disease.

Additionally, for patients who exhibit decreased responsiveness to anti-VEGF and corticosteroid therapeutics, targeting IGF-1 through the use of SST analogs may provide an alternative therapeutic pathway. Improving methods for the local delivery of these SST analogues would prevent systemic side effects and provide a novel treatment option for refractory individuals. In summary, the precise regulation of IGF-1 and the development of targeted therapeutic strategies involving IGF-1, and its downstream molecular targets hold significant potential in advancing the management of DR and DME.

## Figures and Tables

**Figure 1 ijms-26-03961-f001:**
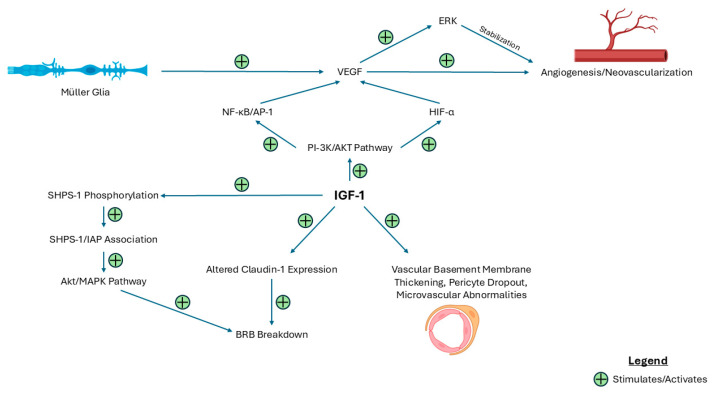
Schematic demonstrating the pathways by which IGF-1 potentiates a pathological response in DR and DME. This figure illustrates the molecular pathways by which IGF-1 contributes to BRB dysfunction, retinal microvascular abnormalities, and neovascularization. IGF-1 activates multiple signaling pathways, including the PI-3K/AKT and SHPS-1 pathways, leading to altered transmembrane protein expression, VEGF induction, and structural vascular changes. Abbreviations: BRB = blood–retinal barrier; ERK = extracellular signal-regulated kinase; IAP = integrin-associated protein; SHPS-1 = Src homology 2 domain-containing protein tyrosine phosphatase substrate 1.

**Figure 2 ijms-26-03961-f002:**
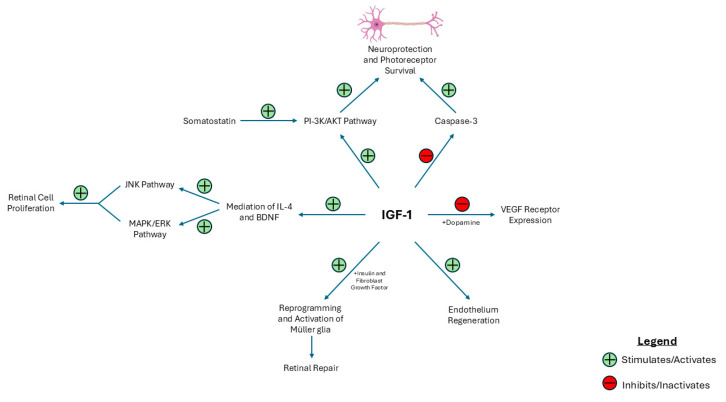
Schematic demonstrating the neuroprotective and regenerative role of IGF-1 in maintaining retinal homeostasis in DR and DME. This figure highlights the beneficial role of IGF-1 in promoting retinal health. IGF-1 supports neuroprotection and photoreceptor survival through activation of the PI-3K/AKT pathway and inhibition of caspase-3-mediated apoptosis and concurrently promotes retinal repair through the action of Müller glia. IGF-1 also directly promotes endothelium regeneration and mediates retinal cell proliferation via MAPK/ERK and JNK signaling. Abbreviations: BDNF (brain-derived neurotrophic factor).

**Figure 3 ijms-26-03961-f003:**
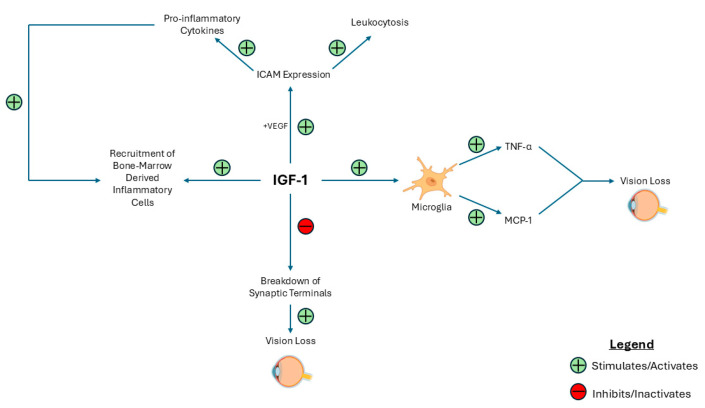
Schematic showing the context-dependent role of IGF-1 in inflammatory processes in DR and DME. This figure demonstrates how IGF-1 contributes to inflammation and vision loss in DR and DME through the recruitment of bone-marrow derived leukocytes and the upregulation of ICAM-1 and microglial activity. Conversely, a complete absence of IGF-1 leads to vision loss through the breakdown of synaptic terminals, indicating its context-dependent role in retinal disease. Abbreviations: ICAM (intracellular adhesion molecule).

**Figure 4 ijms-26-03961-f004:**
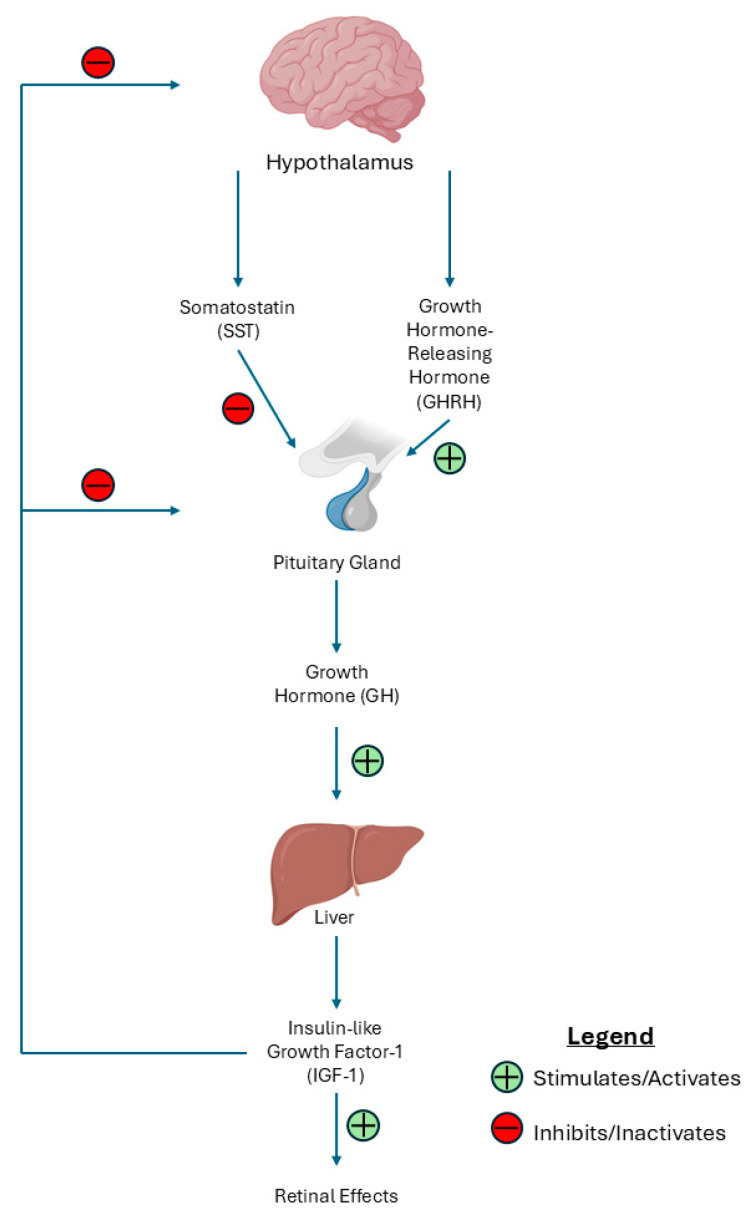
Mechanistic overview of the growth hormone (GH)/insulin-like growth factor 1 (IGF-1) axis leading to downstream retinal effects. This figure illustrates the hypothalamic–pituitary–somatotropic axis, a hormonal pathway regulating growth and retinal homeostasis. The hypothalamus releases growth hormone-releasing hormone (GHRH), which stimulates the pituitary gland to secrete growth hormone (GH). GH then acts on the liver, promoting the synthesis of IGF-1, which mediates several retinal effects. This axis is tightly regulated via negative feedback mechanisms from IGF-1 and GH as well as somatostatin (SST).

**Table 1 ijms-26-03961-t001:** Summary of the key findings and supporting evidence discussed in this review.

Key Finding	Evidence
IGF-1 promotes vascular changes.	IGF-1 upregulates VEGF expression via PI-3K/AKT and JNK/AP-1 pathways and stabilizes neovasculature.
IGF-1 disrupts blood–retinal barrier integrity.	IGF-1 downregulates claudin-1 and increases aberrant activity of AKT and MAPK via SHPS-1 phosphorylation and contributes to structural microvascular abnormalities.
IGF-1 promotes retinal cell survival and repair.	Apoptosis is reduced via modulation of the PI-3K/AKT and caspase-3 pathways. Retinal repair and endothelium regeneration are also promoted via direct effects. Retinal cell proliferation is modulated via IL-4 and BDNF.
IGF-1 drives retinal inflammation.	The upregulation of proinflammatory cytokines such as TNF-α and MCP-1 and the expression of ICAM-1 contributes to leukostasis and microglial activation.
IGF-1 interacts with several signaling molecules.	TGF-β suppresses retinal neovascularization and leukostasis. Estrogen provides neuroprotection via the PI-3K/AKT and MAPK pathways. Dopamine prevents VEGF-1 receptor expression. Angiotensin II promotes the M2 phenotype of microglia.
Clinically, the link between IGF-1 and the progression of DR and DME remains controversial.	Conflicting studies report both positive and negative correlations between serum IGF-1 and DR and DME progression.
Targets of the GH/IGF-1 axis provide potential therapeutic avenues for the management of DR and DME.	Somatostatin (SST) analogues have shown efficacy in the treatment of DR. Novel delivery methods may prevent unwanted side effects.

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
