# Peer review of "The Role and Diagnostic Potential of Insulin-like Growth Factor 1 in Diabetic Retinopathy and Diabetic Macular Edema"

_ijms, 2025, doi:10.3390/ijms26093961_

Round 1

Reviewer 1 Report

Comments and Suggestions for Authors

The authors have summarized the extant literature on ocular effects of IGF-1, a topic to which Dr. Grant has contributed substantially. Overall, the manuscript is well written and considers both ocular and systemic IGF-1 roles. The authors point out remaining questions in the field.

  1. Reference 12 is not available in PubMed and should be replaced.
  2. a 2021 paper by Huber (PMID 34956670) should be cited
  3. a 2019 paper by Raman (PMID 31369407) should be cited
  4. a longitudinal study of serum IGF-1 in T1D (PMID 29663652) should be cited. Throughout the manuscript the emphasis is on increased IGF-1 levels but this study clearly shows that reduced serum IGF-1 is associated with worse outcomes.
  5. The work of Haricot (reference 65) indicates experimentally that intraocular but not systemic IGF-1 causes retinal neovascularization. The authors should comment on the clinical implications of this data, if any.

Author Response

Thank you very much for taking the time to review this manuscript. Please find the detailed responses below and the corresponding revisions/corrections highlighted in red in the re-submitted files. 

1. Reference 12 is not available in PubMed and should be replaced.

The reference has been updated accordingly in the "References" section (line 443).

2. a 2021 paper by Huber (PMID 34956670) should be cited

The results and findings from this study have been summarized and added in line 313.

3. a 2019 paper by Raman (PMID 31369407) should be cited

The findings of this paper have already been cited in the initial submission (citation number 85, line 580).

4. a longitudinal study of serum IGF-1 in T1D (PMID 29663652) should be cited. Throughout the manuscript the emphasis is on increased IGF-1 levels but this study clearly shows that reduced serum IGF-1 is associated with worse outcomes.

The results and findings from this study have been summarized and added in line 324.

5. The work of Haricot (reference 65) indicates experimentally that intraocular but not systemic IGF-1 causes retinal neovascularization. The authors should comment on the clinical implications of this data, if any.

Because this study suggests that intraocular IGF-1 may also contribute to the progression of DR and DME, this has been discussed in the conclusion, stating that local delivery of SST analogues can be used as a therapeutic option, as intraocular IGF-1 levels specifically have also been shown to contribute to the progression of DR and DME. 

Thank you again for your support in reviewing this manuscript.

Reviewer 2 Report

Comments and Suggestions for Authors

The objective of this review was to examine the role of insulin-like growth factor 1 (IGF-1) in the development of diabetic retinopathy (DR) and diabetic macular edema (DME), highlighting its potential use as a marker and therapeutic target. Overall, it is very well conducted and provides valuable information for the area of interest.

Addressing the following aspects could improve the manuscript:

  1. What methodology was used to search for information?
  2. The review focuses on the mechanisms of IGF-1 involved in the progression of DR and DME, without considering the diabetes pathophysiology. It may be important to correlate the mechanisms of this factor in the development of diabetes and this important microvascular complication. Is there any information available on this topic?
  3. The figures are not mentioned in the text.
  4. It would be important to include more specific figure titles and a brief description for better understanding.
  5. Although the review is well described, it would be important to include a table summarizing the main ideas and findings.

Author Response

Thank you very much for taking the time to review this manuscript. Please find the detailed responses below and the corresponding revisions/corrections highlighted in red in the re-submitted files.

1. What methodology was used to search for information?

A PubMed search with relevant keywords was utilized to collect relevant articles. 

2. The review focuses on the mechanisms of IGF-1 involved in the progression of DR and DME, without considering the diabetes pathophysiology. It may be important to correlate the mechanisms of this factor in the development of diabetes and this important microvascular complication. Is there any information available on this topic?

Thank you very much for the suggestion. There are several papers that analyze the role of IGF-1 on diabetes in general. One example of this is the following: PMID: 22374641. While we agree that this would be an interesting topic, we feel that this would be beyond the scope of this review, as we are solely focused on the role and mechanistic potential of IGF-1 specifically on the microvascular complication of DR and DME. 

3. The figures are not mentioned in the text.

The figures have now been mentioned in the text (lines 135, 189, 257, 367).

4. It would be important to include more specific figure titles and a brief description for better understanding.

Titles and a brief description have been added in for each figure (lines 138, 192, 261, 370).

5. Although the review is well described, it would be important to include a table summarizing the main ideas and findings.

Table 1 consolidates the main ideas and findings of the role of IGF-1 specifically (line 419).

Thank you again for your support in reviewing this manuscript.

Round 2

Reviewer 1 Report

Comments and Suggestions for Authors

The authors have generally responded well to the initial comments. However, I find the manuscript to be confusing because of overlapping discussions of the effects of ocular and systemic IGF1. The paper would be greatly strengthened by: 1) a clear description of normal physiological effects of IGF1 within the retina; 2) a succinct description of the ocular effects of increased IGF1; and 3) a clear summary of the systemic effects of decreased and increased serum IGF1.

The text mentions clinical studies of IGF1 therapy and search of clinical trials.gov reveals NCT06881888, "Intranasal Delivery of Octreotide for Treatment of Diabetic Macular Edema," which is not mentioned in the text but Dr. Grant is listed as the PI. A Google patent search reveals she previously held a patent (CA2446101A1) that included IGF1 manipulation via a ribozyme. Does she still have intellectual property on this topic?

Also, the authors should cite PMID38300646.

Author Response

Thank you very much for taking the time to review this manuscript. Please find the detailed responses below and the corresponding revisions highlighted in the re-submitted files.

1. The authors have generally responded well to the initial comments. However, I find the manuscript to be confusing because of overlapping discussions of the effects of ocular and systemic IGF1. The paper would be greatly strengthened by: 1) a clear description of normal physiological effects of IGF1 within the retina; 2) a succinct description of the ocular effects of increased IGF1; and 3) a clear summary of the systemic effects of decreased and increased serum IGF1. The intraocular effects of IGF-1 within the retina have been given from sections 2-4 (lines 54-293). Systemic IGF-1 has been discussed in section 5 now, which focuses on systemic/serum IGF-1 specifically (lines 294-336).   2. The text mentions clinical studies of IGF1 therapy and search of clinical trials.gov reveals NCT06881888, "Intranasal Delivery of Octreotide for Treatment of Diabetic Macular Edema," which is not mentioned in the text but Dr. Grant is listed as the PI. A Google patent search reveals she previously held a patent (CA2446101A1) that included IGF1 manipulation via a ribozyme. Does she still have intellectual property on this topic? The NCT06881888 trial is not yet underway. Additionally, the patent CA2446101A1 was from a very long time ago and nothing commercial has emerged to this day.   3. Also, the authors should cite PMID38300646. This work has now been cited (line 510).    Thank you again for your taking the time to review this paper.  

Round 3

Reviewer 1 Report

Comments and Suggestions for Authors

the authors have addressed my comments